# NETs Are Double-Edged Swords with the Potential to Aggravate or Resolve Periodontal Inflammation

**DOI:** 10.3390/cells9122614

**Published:** 2020-12-05

**Authors:** Ljubomir Vitkov, Bernd Minnich, Jasmin Knopf, Christine Schauer, Matthias Hannig, Martin Herrmann

**Affiliations:** 1Department of Biosciences, Vascular & Exercise Biology Unit, University of Salzburg, 5020 Salzburg, Austria; lvitkov@yahoo.com (L.V.); bernd.minnich@sbg.ac.at (B.M.); 2Clinic of Operative Dentistry, Periodontology and Preventive Dentistry, Saarland University, 66424 Homburg, Germany; 3Department of Internal Medicine 3—Rheumatology and Immunology, Friedrich-Alexander-University Erlangen-Nürnberg (FAU) and Universitätsklinikum Erlangen, 91054 Erlangen, Germany; jasmin.knopf@uk-erlangen.de (J.K.); Christine.Schauer@uk-erlangen.de (C.S.); martin.herrmann@uk-erlangen.de (M.H.)

**Keywords:** NET insufficiency, PMN hyper-responsiveness, ulceration, crevicular occlusion, exaggerated immune response

## Abstract

Periodontitis is a general term for diseases characterised by inflammatory destruction of tooth-supporting tissues, gradual destruction of the marginal periodontal ligament and resorption of alveolar bone. Early-onset periodontitis is due to disturbed neutrophil extracellular trap (NET) formation and clearance. Indeed, mutations that inactivate the cysteine proteases cathepsin C result in the massive periodontal damage seen in patients with deficient NET formation. In contrast, exaggerated NET formation due to polymorphonuclear neutrophil (PMN) hyper-responsiveness drives the pathology of late-onset periodontitis by damaging and ulcerating the gingival epithelium and retarding epithelial healing. Despite the gingival regeneration, periodontitis progression ends with almost complete loss of the periodontal ligament and subsequent tooth loss. Thus, NETs help to maintain periodontal health, and their dysregulation, either insufficiency or surplus, causes heavy periodontal pathology and edentulism.

## 1. Introduction

Neutrophil extracellular traps (NETs) are evolutionary conserved innate immunity structures produced by activated polymorphonuclear neutrophils (PMNs) mainly as a response to pathogen challenge. NETs have a backbone of DNA and are decorated with histones and neutrophil protease, as well as other bactericidal agents, such as lactoferrin, cathepsins and myeloperoxidase (MPO) [1,2,3]. NET formation can be stimulated by activation of neutrophils via receptors for (I) chemokines, (II) cytokines, (III) immune complexes, (IV) pathogen-associated molecular patterns (PAMPs), (V) damage-associated molecular patterns (DAMPs), (VI) C3a or 5a, and (VII) complement C3 and C4 and their derivatives [4]. Many of the mechanisms for NET formation initiated by engagement of the aforementioned receptors are linked to the NADPH oxidase (NOX) machinery, but NOX-independent processes have also been described [5,6,7]. Upon activation, the azurophilic granular proteins NE and MPO translocate to the nucleus to promote nuclear and chromatin decondensation [8]. Histone hypercitrullination by peptidylarginine deiminase 4 (PADI4) also contributes [9]. However, NE- and PADI4-independent pathways have also been reported [10,11].

Depending on the context, NETs are known to exert either pro- or anti-inflammatory effects. NETs induce the inflammasome [12], type I interferons and pro-inflammatory cytokines; damage the endothelium [13]; and can occlude ducts in various organs, thereby promoting organ damage [14,15,16]. Furthermore, disbalanced NET formation carries the risk of developing periodontal pathologies.

In areas of low neutrophil densities (e.g., in blood or during early stages of inflammation), individual neutrophils form NETs and trap pathogens in an inflammatory manner, releasing their load of proinflammatory mediators (e.g., cytokines, chemokines). Consequently, further neutrophils are attracted, and the density of neutrophils increases. In areas of high cell densities, the NET-forming neutrophils clump together and form felted aggregates (aggNETs). The latter initially trap and finally degrade several inflammatory mediators. The NET structures exert anti-inflammatory effects leading to downregulation of inflammatory responses by promoting the trapping and cleavage of pro-inflammatory mediators through NET-bound proteases [17,18,19]. This process initiates the resolution of inflammation.

Periodontal inflammation starts as gingival inflammation, which is completely reversible. Periodontitis is either early onset (earlier denoted aggressive) or late-onset (earlier denoted chronic). Despite the common morphological destruction of the periodontium, these disease entities differ in their inflammatory response. The early onset periodontitis affects both human dentitions and is a consequence of genetic defects [20] mostly concerning PMNs. Impaired neutrophil extracellular trap (NET) formation, resulting from e.g., mutations that inactivate cathepsin C [21] or neutrophils elastase (NE) [22], are accompanied by severe forms of periodontitis. In contrast, late-onset periodontitis predominantly emerges in individuals with an age above 35 years [23]. It is characterised by a hyper-responsiveness of the PMN [24,25,26,27,28]. Late-onset periodontitis is a heritable [29] inflammatory age-related disease of tooth supporting tissues [30], characterised by resorption of periodontal ligament and alveolar bone. With the resorption of periodontal ligament, the marginal gingival sulcus deepens, and the so-called periodontal pocket forms, a deep narrow space enclosed between the gingival epithelium and the dental root. The denudate root is frequently covered with subgingival concrements and/or dental plaque and is emerged in gingival crevicular fluid, a blood plasma transudate [31] rich in activated neutrophils and NETs [32,33] (Figure 1). Late-onset periodontitis progression takes place through recurrent bursts of destruction followed by varying periods of stagnation. This temporally and spatially haphazard mode of disease progression has been named the random burst model [34,35,36]. Periodontitis progression ends with tooth exfoliation [37].

Currently, the prevailing opinion is that inflammatory bone resorption is the main driving force of alveolar bone destruction in late-onset periodontitis [38,39,40,41]. Osteoclast differentiation and activation are promoted by the binding of receptor activator of nuclear factor kappa-Β ligand (RANKL) (expressed by osteoblasts and activated T-cells and B-cells) to RANK on osteoclast precursors; osteoprotegerin acts to block the RANKL–RANK interaction and restrains osteoclastogenesis. It has been assumed that Th17 cells, activated by oral pathogens, induce bone destruction via production of RANKL and subsequent osteoclast proliferation. Following the hypothesis of the crucial IL17 role in late-onset periodontitis [40,41] and considering the fact that the implant–gingiva attachment is not tighter than the tooth–gingiva one, dental implants in patients with late-onset periodontitis should also be prone to inflammatory osteolysis to the same extent, but this is not the case. Dental implants in patients with late-onset periodontitis have the same or only slightly reduced surviving rate as compared to patients without late-onset periodontitis [42]. The lack of positive effects of bisphosphonates on late-onset periodontitis in humans [43] indicates that the inflammatory osteolysis [38,39,40,41] cannot be the main reason for late-onset periodontitis. All of these findings cast doubt on the idea that Th17-induced osteolysis be the primary driving force of bone resorption in late-onset periodontitis.

Many epidemiological data demonstrate that both cells and molecules of the innate and adaptive immune response are adversely impacted by aging, yielding a reasonable tenet that the increased periodontitis noted in aging populations is reflective of the age-associated immune dysregulation [44,45]. Thus, an increased basal level of NF-κB activation has been reported in dendritic cells (DCs) from aged subjects [46]. The epithelial cells in the mucosa secrete factors, such as retinoic acid and TGF-β, which act upon DCs to induce tolerance to prevent response against harmless antigens and commensal microbiome [47,48,49]. DCs from aged subjects display impaired response to retinoic acid and are deficient in inducing T regulatory cells for tolerance [50]. The basal level of activation of DCs from aged subjects leads to low-level secretion of pro-inflammatory cytokines, which activates the epithelium even in the absence of infection [51]. Thus, PMN hyper-responsiveness in late-onset periodontitis PMN [24,25,26,27,28] and loss of tolerance [52], i.e., an exaggerated immune response, appears reasonable as a main cause of late-onset periodontitis [53,54,55,56,57].

## 2. NET Formation and Aggregation in Late-Onset Periodontitis

Canonical NETs, adequate to bacterial challenge, are a main protector of periodontal tissues, and insufficient NETs cause severe early-onset periodontitis [56]. However, exaggerated NET formation may be responsible for tissue damages in late-onset periodontitis. The exaggerated PMN response in late-onset periodontitis may have various causes: one of them is the hyper-responsiveness of PMNs in late-onset periodontitis [24,25,26,28]. Additionally, altering the mucosal PMNs due to the mucosal transmigration takes place, particularly characterised by delayed PMN apoptosis [58,59], as evident by transcriptome changes in oral PMNs [60]. NETs form exclusively on the surface of the crevicular (pocket) epithelium [32]. Deep pockets have a resting volume of up to 1.5 µL and also gingival crevicular fluid flow rates of up to 44 µL/h [34], i.e., the gingival crevicular fluid has a very short dwelling time of nearly 2 min within the crevice. However, the time of PMN adhesion to the epithelial surface may prolong the PMN dwelling time within the crevice. Nevertheless, this time is too short, as no PMNs accumulate on the crevicular surface in late-onset periodontitis [32,61]. However, the canonical NET formation requires at least 2 h [2]. The only type of NET formation in a few minutes is the bicarbonate-induced non-canonical NET formation [15].

Cystic fibrosis transmembrane conductance regulator (CFTR) is expressed in a number of epithelia in humans, including the gingival epithelium. Moreover, CFTR expression in gingival epithelial from periodontitis biopsies is strongly elevated and even present in the gingival connective tissue [62]. Degradation of HCO^3−^ by crevicular carbonic anhydrases (CAs) causes alkalinisation. Indeed, CA-1 is regularly located in PMNs [63], some epithelia [64] and both healthy [65] and CPD-affected gingiva [66]. In particular, elevated CA-1 of gingival crevicular fluid in late-onset periodontitis [65,66,67,68] suggests a connection to elevated pH of gingival crevicular fluid. The pH of gingival crevicular fluid varies in late-onset periodontitis from 6.9 up to 8.7 due to the high bicarbonate concentration [69,70], but direct measurement of pH within the gingival pocket has yielded a somewhat lower range of 6.35–8.10 [71]. Gingivitis does not appear to be associated with major changes in crevicular pH, and the mean crevicular pH is near 7.0, but in about 7% of periodontitis individuals, there is at least one site with a pH between 7.5 and 7.8 [71]. Bicarbonates above 37 mM and pH above 7.55 cause extremely quick NET formation, namely non-canonical NETs [72], as in the case of pancreatitis [15]. Thus, a bicarbonate non-canonical NET formation may be expected in the alkaline gingival crevicular fluid of late-onset periodontitis. The elevated gingival CFTR expression in late-onset periodontitis [62] is an important requirement for the exaggerated NET formation, as it enables the bicarbonate enrichment and alkalosis of gingival crevicular fluid. Cystic fibrosis (CF) patients have mutated CFTR and completely lack the ability to secrete HCO^3−^. However, data concerning the late-onset periodontitis in CF patients are inconclusive [73] due to fact that the life expectancy of CF patients is below 50 years, and this period is too short for the development of late-onset periodontitis. However, gingivitis, the precursor of late-onset periodontitis, is significantly lower in adult CF patients as compared to healthy controls [73,74], and the periodontal risk is at a low level [75]. Particularly, aggregated NETs (aggNETs) [18] have also been reported [32] in purulent gingival crevicular fluid, but the final stage of bicarbonate-induced NETosis [72] has never been reported, probably due to the short dwelling of PMNs within the gingival periodontal crevice.

Exaggerated NETs and, hence, oversupply of proteases in the crevice are major tissue-damaging factors in late-onset periodontitis.

## 3. Periodontal Damage by PMNs, NETs and NET Degradation Products

In periodontal tissues and crevices, both serine and metalloproteases are almost exclusively delivered by PMNs [76,77,78]. The NETs may dissolve the basal lamina [79] and damage or even kill the epithelial cells [80] causing epithelial ulceration. Indeed, increased gingival crevicular fluid levels of laminin-332 [81,82,83], PMN proteases [76,77,78] and the epithelial ulceration in late-onset periodontitis [84,85] have been reported. Epithelial ulceration may be assisted by periodontal pathogens [86] and enables an unimpeded influx of NET-derived proteases into the periodontal ligament. Patients with late-onset periodontitis are regarded as being immunised with periodontal pathogens [87]. Immunisation against bacterial components and the subsequent formation of immune complexes induces PMN infiltration, NET formation [88,89] and thus precipitates periodontitis [87,90]. Smoking further aggravates late-onset periodontitis [91,92,93,94]. Cigarette smoke fosters NET formation [95] and PMN-derived proteases, which are responsible for the connective tissue breakdown [96]. In both mentioned cases of external damaging factors, PMNs and NETs are key mediators of the periodontal tissue injury.

Regardless of how destruction of the junctional epithelium and the underlying periodontal ligament takes place, it inevitably results in long junctional epithelium formation [87] (Figure 1). After destruction of marginal periodontal ligament, keratinocyte proliferation provides the seal of the connective tissue wound, namely, the long junctional epithelium, which replaces the destroyed connective tissues. As a result, the marginal periodontal ligament is gradually destroyed with each exacerbation [34,35,36] up to exfoliation of the affected tooth.

Alveolar bone resorption follows the loss of periodontal ligament, as demonstrated in cases post extraction [97,98] and in animal models [99,100,101]. Thus, bone remodelling rather than inflammatory osteolysis appears to be responsible for the alveolar bone resorption in late-onset periodontitis.

## 4. Crevicular Occlusion by NETs

The periodontal crevice, i.e., the space between dental root and the crevicular gingiva, is filled with gingival crevicular fluid. The length of the crevice increases with deepening of the periodontal pocket and can even rich the root apex. Gingival crevicular fluid is continuously produced by the gingival epithelium and drained into the oral cavity. Thus, the crevice may be viewed as a duct, which evacuates the gingival crevicular fluid along with bacteria, pathogen-associated molecular patterns (PAMPs) and damage-associated molecular patterns (DAMPs) from the crevicular bottom into the oral cavity (Figure 1). The quantity of gingival crevicular fluid and its density, i.e., content of NETs, depend on whether remission or exacerbation of late-onset periodontitis takes place. In most heavy cases of exacerbation, purulent exudate is produced [102]. This exudate is rich in aggNETs and extremely viscous [32]. The extracellular DNA excess of aggNETs cause increased viscosity [103,104] of gingival crevicular fluid and, as a result, obstruction of or even possibly occlusion of gingival crevicular fluid evacuation out of the crevice. Such a mechanism of a duct occlusion by bicarbonate-induced NETs has been reported in pancreatitis [15,72]. Marginal occlusion of the pocket leads to gingival crevicular fluid retention, abscess formation in the pocket and its extension into the surrounding periodontal tissues [105,106,107]. In a few cases, pH of saliva in the quiescent state is 7.5 or somewhat higher, but does not reach 8.0 [108]. However, as a reaction to oesophageal charges, salivary pH rises up to 8.02 (a 3.3-fold increase), and the salivary bicarbonate concentration also rises, showing a 3.7-fold increase [109,110]. Thus, in certain cases, high salivary pH might critically accelerate NET formation in the marginal periodontal crevice and contribute to its occlusion. Furthermore, salivary sialyl Lewis^X^ induces NETs within 15 min, thus possibly contributing to the occlusion of the marginal periodontal crevice [111]. Generally, two modes of periodontal abscess development in periodontitis patients have been reported: (i) the unprompted one in untreated periodontitis and (ii) after periodontitis treatment [106]. The unprompted periodontal abscess might be a result of occlusion by bicarbonate NETs, whereas after periodontitis treatment, an additional mechanism may also come into question. Surgical wounds in marginal areas of the periodontal pocket heal by long junctional epithelium formation [87] in three days, even in the absence of stromal healing [112]. This ability is due to the high activation of the oral epithelium showing upregulation of genes linked to epithelial and immune cell migration, as well as increased proliferation, as demonstrated by analysis of the transcriptome of the oral epithelium [112]. The capability of the gingival epithelium to “reattach” to the root surface by long junctional epithelium formation is routinely used in the modified Widman flap surgical technique [113], where, however, the whole crevicular epithelium is removed. Healing the marginal areas of the periodontal pocket comprises the epithelial attachment to the root surface and may lead to occlusion of deeper parts of the periodontal pocket (Figure 2). The contaminated pocket epithelium below the epithelium reattachment continues to produce gingival crevicular fluid, which cannot be eliminated by draining. The marginal crevicular occlusion may further increase the PMN transmigration into the crevice and ultimately results into periodontal abscess formation.

Independent of how the crevice occludes, the pocket epithelium below the place of occlusion sustains PMN transmigration and NET formation, and this retention yields a periodontal abscess.

## 5. NETs in Gingival Ulceration Healing

The pocket epithelium is characterised by the tendency to reveal micro-ulcerations [84]. Epithelium ulcerations increase the chance for invasion of microorganisms and the penetration of their PAMPs into the soft connective tissue, thereby aggravating the course of periodontitis. However, transitory bacteraemia is not pathognomonic for periodontitis. Mechanically caused gingival micro-ulcerations result in transitory bacteraemia lasting minutes [114,115,116]. Similarly, the skin, which has a considerably larger surface, undergoes frequently multitudes of micro-traumas such as insect bites or minor skin scratches, which also results in transitory bacteraemia [117]. A main task of NETs is to limit the bacterial spreading [1]. However, in late-onset periodontitis [118,119] and in experimental periodontitis [40], i.e., in a heavily contaminated pocket epithelium, crevicular NETs are overwhelmed and appear unable to completely prevent periodontal pathogen dissemination to inner organs, a finding demonstrating the limitation of NETs to prevent bacterial spreading in a heavily contaminated environment. However, exaggerated NET formation suppresses wound healing [120]. Three findings indicate the suppressive effect of exaggerated NET formation in late-onset periodontitis: (i) increased NET degradation has been observed in late-onset periodontitis as a consequence of periodontal therapy [121]; (ii) the correlation between diabetes and late-onset periodontitis [122,123] appears to be a consequence of increased NET formation in diabetes [120]; (iii) cigarette smoke, a factor aggravating the course of periodontitis, fosters NET overproduction [95].

Either suppression of exaggerated NET formation or NET lysis by DNase can ameliorate the negative effects of NETs on wound healing [120]. The healing of gingival ulcerations in late-onset periodontitis by attenuation of NET formation, e.g. through NET lysis, opens a new perspective to amplify the late-onset periodontitis therapy.

## 6. NET-Driven Periodontal Inflammation

PMN hyper-responsiveness [24,25,26,27,28], delayed PMN apoptosis [124], immune-senescence [46,50,125] and PAMP overload, i.e., dental plaque accumulation, dominate the immune response in late-onset periodontitis [53,54,55,56,57]. Once late-onset periodontitis occurs, peripheral PMN hyper-responsiveness is evident, which does not disappear even after successful periodontal therapy [126]. The PMN hyper-responsiveness results in exaggerated NET formation and even in epithelial damages including ulceration [120] and, thus, sustains and aggravates periodontal inflammations. The oral epithelium healing mechanism cannot compensate the damages due to the exaggerated NET formation and is incapable of providing epithelial repair. As consequence, heavy inflammatory infiltration of the connective tissue and periodontal ligament damaging take place. The healing of damaged periodontal ligaments occurs via replacement with the junctional epithelium [127]. Due to the high proliferation rate of oral epithelial cells, after just a few days, the large part of the periodontal wound would inevitably be delimitated by the epithelium [128], even in the absence of stromal healing [112]. This phenomenon is denoted either apical migration of the junctional epithelium, or long junctional epithelium formation. This healing through repair reduces the periodontal ligament surface and subsequently the load-bearing capacity of the affected tooth. This results in periodontium overburdening, which causes trauma-induced inflammation. Thus, even when the gingival inflammation is in a state of remission [34,35,36], moderate bacteria challenges or mechanical overloading may inflame the inflammation anew.

The PMN hyper-responsiveness is considered the main pathogenic factor in localised aggressive periodontitis, a subtype of late-onset periodontitis [126], and is also responsible for overproduction of NETs. Exaggerated NETs cause epithelial cytotoxicity [126,129] and liberate proteases, damaging even the basal lamina [79]. This clearly indicates the destructive role of exaggerated NET formation in periodontitis. Clinically, the topical application of prednisolone alone [130] into the periodontal pocket in cases of periodontitis exacerbation is sufficient for a temporary resolution of periodontitis inflammation, or at least for its attenuation. Similarly, topical application of resolvin E1 in an animal model caused resolution of inflammation via PMN depriming [130]. These findings clearly indicate that the bacterial challenge in periodontitis plays a secondary role.

The exaggerated immune response in late-onset periodontitis, in particular, the exaggerated NET formation, appears to be responsible for tissue damages, impairs tissue regeneration and causes non-resolution of inflammation.

## 7. Role of NETs in the Resolution of Gingival Inflammation

Periodontal NETs have been reported on for over a decade [32], but their role in the resolution of gingival and periodontal inflammation remains largely unexplored due to technical difficulties in the examination of periodontal NETs in both humans and animal models, as well as the lack of interest on the side of periodontal societies. NET indispensability for periodontal health is nevertheless evident in patients with certain gene defects that are inevitably concomitant with both deficient NET formation and early-onset periodontitis [57]. The early-onset periodontitis is resolved only with the teeth loss [21,22]. These findings indicate the crucial role of canonical NETs in the resolution of gingival inflammation.

The inner mucous layer of the mucosal epithelia, where goblet cells are located, shields and aggregates bacterial pathogens [131]. Gingival epithelia lack the mucus layer, and its task has been suggested to be taken over by NETs [57]. The role of NETs in maintaining periodontal health and resolution of periodontal inflammation is comprehensible through analogy to other mucosal surface-lacking goblet cells, e.g., the cornea. Corneal NETs render a “dead zone” for bacteria and prevent tissue invasion by bacteria [132]. Indeed, the NET ability to block off and aggregate bacteria and nano-particles is beyond doubt [133,134].

Late-onset periodontitis progression is characterised by recurrent bursts of destruction, followed by varying periods of stagnation, the so-called random burst model [34,35,36]. The mechanism of this “spontaneous” switch from exacerbated inflammation to resolution of inflammation remains obscure. However, aggNETs, which are common in purulent periodontitis [32], are able to resolve inflammation through the degradation of inflammatory cytokines and mediators [18], as well as by detoxifying the extracellular histones [19]. Thus, aggNETs might be a crucial factor for the resolution of exacerbated periodontal inflammation.

Further in vivo and clinical examinations on crevicular aggNETs are needed to clarify their role in the pathology of periodontitis.

## 8. Treatment Approaches of Periodontitis Based on NET Suppression

The PMN hyper-responsiveness [24,25,26,27,28] in late-onset periodontitis appears to be the reference point for the understanding the periodontal pathology, but its cause remains obscure. One reason for PMN hyper-responsiveness might be the lipopolysaccharide (LPS) overload, due to dental plaque accumulation as a consequence of the consumption of carbohydrate-rich thermally processed food. Thus, Alaska Natives on their traditional diet, i.e., raw meat, have low rate of periodontitis [135]. In urban societies, temporary reduction of LPS load is achieved by tooth brushing and antibiotic application. In addition, PMN hyper-responsiveness is a common feature of diabetes, rheumatic arthritis, and other inflammatory systemic diseases [56,136], which are frequently concomitant with late-onset of periodontitis. PMN hyper-responsiveness in late-onset periodontitis results in exaggerated NET formation, which is responsible for gingival damages. Consequently, either lowing the PMN hyper-responsiveness or attenuation of exaggerated NET formation might be a useful treatment approach in late-onset periodontitis. The unawareness of the reason for PMN hyper-responsiveness rather highlights the application of topical anti-inflammatory substances, as they have minimal adverse effects or none at all. Many substances acting on NETs promise new treatment approaches in periodontitis treatment. They either attenuate NET formation or diminish the deteriorating effects of exaggerated NETs and belong to different classes of medicaments:
(i)Inhibitors of Toll-like receptors: a promising candidate is berberine, which provides significant protection against LPS-induced mucosal injury in mice, via inhibiting the TLR4-nuclear factor κB-MIP-2 pathway and decreasing neutrophil infiltration [137].(ii)Another promising approach is the topical treatment with specialised pro-resolving mediators, which stun PMNs, e.g., resolvins, lipoxin A4 etc. [138].(iii)Topically applied prednisolone: corticosteroids downregulate NETs [139] and are efficient and prevalent medication in periodontology [130].(iv)Doxycycline and its derivates are metalloproteinase inhibitors and efficiently downregulate the NET metalloproteinases in a sub-antimicrobial dose applied as a systemic adjunctive or a topical therapy [43].(v)Systemically applied nonsteroidal anti-inflammatory drugs, which downregulate NETs [140].

The cognition that exaggerated NETs in late-onset periodontitis are a deteriorating factor introduces new possibilities for treatment approaches.

## 9. Conclusions

Canonical NET formation appears an indispensable protective mechanism of gingiva, as NET insufficiency causes early-onset periodontitis. In contrast, the exaggerated NET response in late-onset periodontitis is tissue damaging. The exaggerated NET formation is due to genetic predisposition (which occurs age-dependently), PMN hyper-responsiveness and apoptosis resistance, as well as bacterial overload (dental plaque accumulation) due to the consumption of carbohydrate-rich thermally processed food. Understanding the role of NETs in periodontitis pathology enables new analytical treatment approaches supporting the empirics.

## Figures and Tables

**Figure 1 cells-09-02614-f001:**
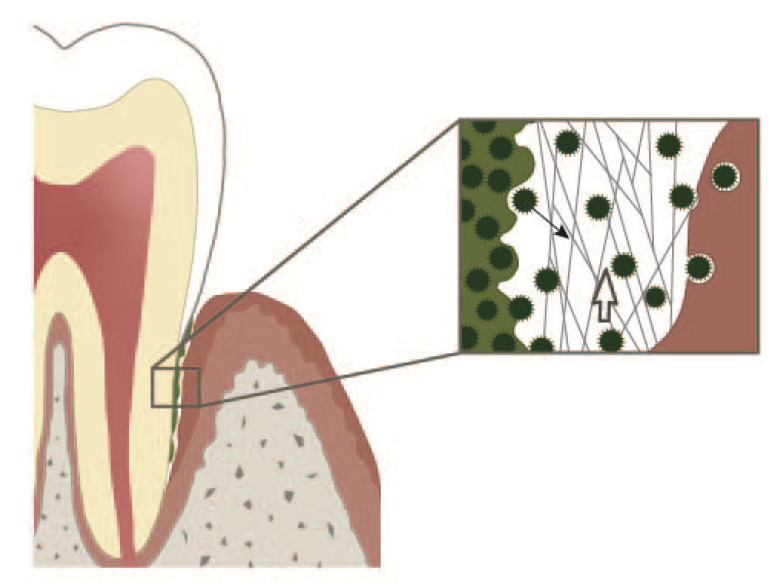
Periodontal crevice. On the left: subgingival plaque. On the right: an epithelial cell with an adherent bacterium and bacteria in different stages of internalisation. In the middle: periodontal crevice. Neutrophil extracellular trap (NETs) build a 3D web work protecting the gingiva from bacteria dispersing out of the subgingival plaque. NETs together with the entrapped bacteria are continuously pushed into the oral cavity by the crevicular exudate outflow. Open white arrow: direction of the crevicular exudate outflow. Solid black arrow: a bacterium dispersing from the subgingival plaque.

**Figure 2 cells-09-02614-f002:**
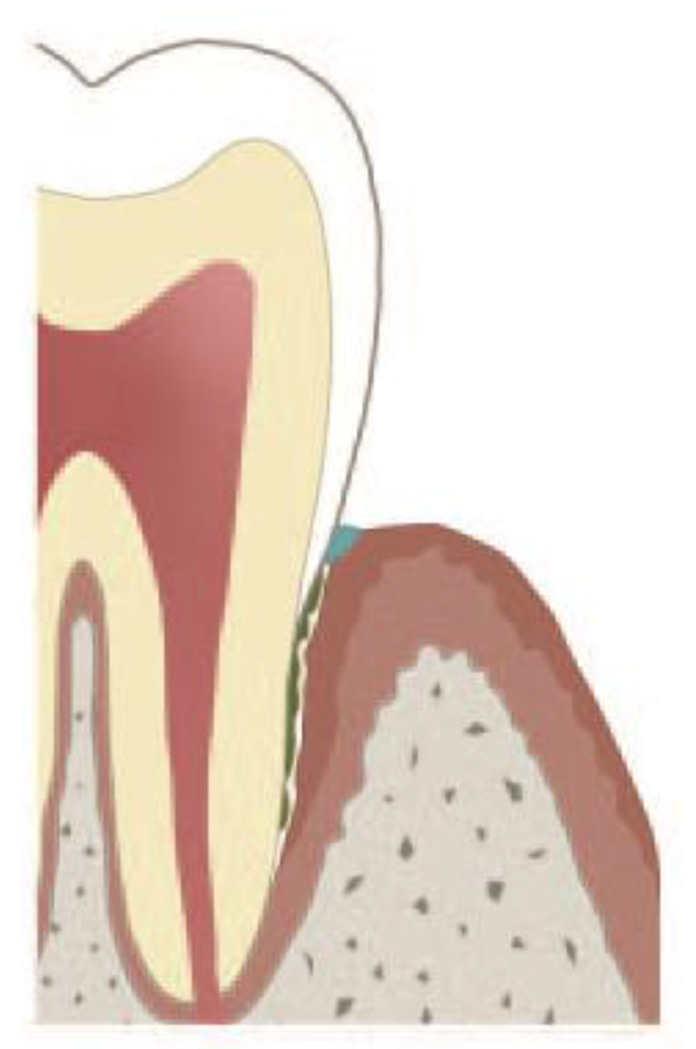
Occlusion of marginal pocket either by aggregated NETs (aggNETs), or by attachment of the marginal epithelium to the root. The underlying pocket epithelium continues to secrete gingival crevicular fluid, and polymorphonuclear neutrophils (PMNs) continue to transmigrate. The exact nature of the plug (depicted blue) has not been clarified yet [87,113].

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
