# Peer review of "NETs Are Double-Edged Swords with the Potential to Aggravate or Resolve Periodontal Inflammation"

_cells, 2020, doi:10.3390/cells9122614_

Round 1

Reviewer 1 Report

The title in unclear: NET's contribution?

In the abstract (line 21), also in another parts of the manuscript, please write first the meaning of the abbreviation. i.e. polymorphonuclear neutrophils (PMN)

Line 41: proteins NE?

Line 51: NETting?

In the abstract: Is it appropriate to mention Papillon-Lefèvre-Syndrome? It is no longer found anywhere in the text!

Page 2, line 74: tooth exfoliation?

Figure 1 and Figure 2. Please mention the reference for the pictures.

Conclusion: Please, mention the practical implications for clinicians, also.

Author Response

The title in unclear: NET's contribution? The title was changed

In the abstract (line 21), also in another parts of the manuscript, please write first the meaning of the abbreviation. i.e. polymorphonuclear neutrophils (PMN) the suggested correction was made.

Line 41: proteins NE? The phrase was reformulated.

Line 51: NETting? The term was changed into NET forming.

In the abstract: Is it appropriate to mention Papillon-Lefèvre-Syndrome? It is no longer found anywhere in the text! The term was replaced with “deficient NET formation”

Page 2, line 74: tooth exfoliation? The term is common – so deciduous teeth exfoliate, there is no other term in standard English. Teeth are evolutionary modified scales and permanently exfoliate (as the scales do) and new ones erupt in all teethed animal except for mammals, which retain lifelong most of their permanent teeth. Nevertheless, I removed the term from the abstract and inserted a reference into the text, in order all non-native English speakers to be able to look the term up.

Figure 1 and Figure 2. Please mention the reference for the pictures. They were already mentioned in the text, but as abbreviations “fig.”, so I corrected them into “figure”.

Conclusion: Please, mention the practical implications for clinicians, also. Done as suggested.

Reviewer 2 Report

Topic of this manuscript is interesting and up to date. I recommend it for publication without any major revisions. English language and style require minoe check.

Author Response

Minor check of English language and style was done as suggested.